# Caveolin-1-Derived Peptide Reduces ER Stress and Enhances Gelatinolytic Activity in IPF Fibroblasts

**DOI:** 10.3390/ijms23063316

**Published:** 2022-03-18

**Authors:** Satoshi Komatsu, Liang Fan, Steven Idell, Sreerama Shetty, Mitsuo Ikebe

**Affiliations:** Department of Cellular and Molecular Biology, The University of Texas at Tyler Health Science Center, Tyler, TX 75708, USA; satoshi.komatsu@uthct.edu (S.K.); liang.fan@uthct.edu (L.F.); steven.idell@uthct.edu (S.I.); sreerama.shetty@uthct.edu (S.S.)

**Keywords:** idiopathic pulmonary fibrosis, endoplasmic reticulum stress, caveolin-1 scaffolding domain peptide, matrix metalloproteinases

## Abstract

Idiopathic pulmonary fibrosis (IPF) is a fatal disease characterized by an excess deposition of extracellular matrix in the pulmonary interstitium. Caveolin-1 scaffolding domain peptide (CSP) has been found to mitigate pulmonary fibrosis in several animal models. However, its pathophysiological role in IPF is obscure, and it remains critical to understand the mechanism by which CSP protects against pulmonary fibrosis. We first studied the delivery of CSP into cells and found that it is internalized and accumulated in the Endoplasmic Reticulum (ER). Furthermore, CSP reduced ER stress via suppression of inositol requiring enzyme1α (IRE1α) in transforming growth factor β (TGFβ)-treated human IPF lung fibroblasts (hIPF-Lfs). Moreover, we found that CSP enhanced the gelatinolytic activity of TGFβ-treated hIPF-Lfs. The IRE1α inhibitor; 4µ8C also augmented the gelatinolytic activity of TGFβ-treated hIPF-Lfs, supporting the concept that CSP induced inhibition of the IRE1α pathway. Furthermore, CSP significantly elevated expression of MMPs in TGFβ-treated hIPF-Lfs, but conversely decreased the secretion of collagen 1. Similar results were observed in two preclinical murine models of PF, bleomycin (BLM)- and adenovirus expressing constitutively active TGFβ (Ad-TGFβ)-induced PF. Our findings provide new insights into the mechanism by which lung fibroblasts contribute to CSP dependent protection against lung fibrosis.

## 1. Introduction

Idiopathic pulmonary fibrosis (IPF) is a fatal chronic disease that affects between 13,000 and 200,000 individuals and results in 50,000 deaths each year in the United States [1]. There is currently no cure for IPF, and the median survival is less than 3 years after diagnosis [2,3]. Oral administration of either pirfenidone or nintedanib has been reported to slow the progression of IPF, but it is not curative [4,5]. Currently, a new compound, adelmidrol, has reported to be a candidate as a therapeutic approach in the treatment of pulmonary fibrosis (PF) by using bleomycin-induced pulmonary fibrosis in mice [6]. Hence, there remains an urgent need for effective anti-fibrosis compounds that can limit the progression of PF as well as drug-related side effects.

Activation and phenotypic changes of lung fibroblasts to myofibroblasts are believed to be linked to the development of interstitial lung diseases, including IPF [2,7]. In multiple mouse models of experimentally induced lung fibrosis, caveolin-1 scaffolding domain peptide (CSP) has been shown to alleviate PF [8,9,10]. Importantly, CSP appeared to resolve PF even in relatively advanced stages of disease [9], suggesting that this peptide represents a potential candidate for the treatment of patients with IPF.

CSP has been shown to reduce the expression of extracellular matrix (ECM) proteins, including collagen and fibronectin, in human IPF lung fibroblasts isolated from the lung tissues from patients with IPF (hIPF-Lfs) and lung fibroblasts (Lfs) from mice with established PF [9]. In addition to IPF, CSP has also been reported the beneficial effects on scleroderma, cardiac, liver, and renal fibrosis [10,11,12,13,14,15]. However, it is still unclear whether CSP is involved in the degradation of existing ECM in fibrotic lungs and if so, how CSP can facilitate clearance of deposited ECM.

The unfolded protein response (UPR)/endoplasmic reticulum (ER) stress pathway consists of three ER transmembrane proteins, PKR-like ER kinase (PERK), activating transcription factor 6 (ATF6) and inositol requiring enzyme1α (IRE1α) [16,17,18,19,20,21]. Current studies have shown that UPR activation is associated with the development and progression of PF [16,17,20,22]. During the development and progression of IPF, ER stress affects several different cell types in the lungs, including fibroblasts, alveolar macrophages, and alveolar epithelial cells (AEC) [20,21]. Transforming growth factor β (TGFβ) has been established as a transducer of fibroblasts into myofibroblasts as well as PF [23]. Halayko and colleagues showed that an IRE1α inhibitor protected against TGFβ-induced myofibroblast phenoconversion [17]. Interestingly, this feature was only observed for hIPF-Lfs, but not for Lfs from non-IPF donors. In addition, ER stress is also suggested to contribute to AEC apoptosis and M2 polarization macrophages [20,21].

In the present study, we investigated a novel mechanism by which CSP clears existing ECM deposits, which alleviates the progression of IPF in concert with previously reported antifibrotic effects attributed to this peptide [8,9,10]. We found that CSP is accumulated in the ER under ER stress in TGFβ-treated hIPF-Lfs. We also observed that CSP mitigated the ER stress via suppression of IRE1α in TGFβ-treated hIPF-Lfs. Interestingly, inhibition of IRE1α enhanced the gelatinolytic activity of hIPF-Lfs. The process involved upregulation of matrix metalloproteinases (MMPs), and conversely reduced the secretion of collagen 1. These results demonstrate, for the first time, that CSP may reduce accumulated ECM by activating ECM degradation in addition to inhibition of ECM secretion. Our study provides a new insight into the mechanism of CSP-mediated resolution of PF.

## 2. Results

### 2.1. Accumulation of CSP in ER in Lung Fibroblasts from Patients with IPF

To understand the mechanism of CSP-mediated resolution of existing PF, it is important to determine the targeting sub-cellular organelles of CSP. To address this question, we first monitored the delivery and destination of CSP into primary hIPF-Lfs as they are primary effector cells. CSP was tagged with the fluorescent moiety; tetramethylrhodamine (TMR) via N-terminal cysteine. Distribution of CSP-TMR was monitored with a confocal microscope. Serum-starved hIPF-Lfs were stimulated with 2.5 ng/mL TGFβ in the presence of 10 µM CSP-TMR for 48 h. Since caveolin-1 localizes to multiple cell membranes such as the plasma membrane, cell organelle membranes, and vesicle membranes [24], the localization of CSP-TMR in hIPF-Lfs was visualized along with several marker proteins that specifically recognize each cell organelle. These include the ER, Golgi, early endosomes, and mitochondria. As shown in Figure 1, CSP-TMR was incorporated into the cells and localized within the hIPF-Lfs. It should be noted that CSP did not stay at the hIPF-Lf membrane surface for a prolonged period of time and readily internalized into the cells. To visualize the unique localization of CSP-TMR, we performed colocalization analysis with various marker proteins. Among the organelle specific marker proteins tested, we found that CSP-TMR clearly colocalized with calnexin, an ER marker (Figure 1A), but not the marker proteins of other organelles (Figure 1B). These findings suggested that the accumulation of CSP in ER of hIPF-Lfs may lead to modification of ER function.

### 2.2. TGFβ-Induced ER Stress in hIPF-Lfs and in the Fibrotic Lung

Because ER was identified as a main target of CSP in cells, we next studied the effect of CSP on ER function. Since ER stress has been shown to be associated with the pathogenesis and progression of IPF [20,21], we hypothesized that CSP might affect ER stress, which could mitigate PF. To address this postulate, we examined the effect of CSP on the expression of ER stress marker proteins using specific antibodies. Serum-starved hIPF-Lfs were stimulated with 2.5 ng/mL TGFβ in the presence or absence of 20 µM CSP for 48 h, and then subjected to Western blot analysis. As shown in Figure 2, TGF-β induced UPR marker proteins, especially IRE1α and BiP in hIPF-Lfs, consistent with a previous report [17]. Intriguingly, CSP treatment reduced both IRE1α and BiP expression, which is otherwise induced in TGFβ-treated hIPF-Lfs. The result suggests that CSP may be involved in the regulation of ER stress pathways.

We next asked whether ER stress is induced by TGFβ and mitigated by CSP in vivo by using a preclinical mouse model of PF. To address this question, we studied the effect of CSP on recruitment of hIPF-Lfs, ECM deposition, and development of ER stress in Adenovirus expressing constitutively active TGFβ (Ad-TGFβ)-induced PF mouse lung tissue. Consistent with our previous report [9] and others [25,26], Ad-TGFβ induced PF with extensive deposition of collagen and other ECM proteins. Lungs of mice transduced with Ad-TGFβ also showed increased αSMA, a fibrotic fibroblasts marker, indicating recruitment of myofibroblasts (Figure 3B). Consistent with the immunoblot analysis of hIPF-Lfs (Figure 2), lung sections of mice exposed to Ad-TGFβ showed increased fluorescent signals for BiP in αSMA positive cells (Figure 3B, BiP: red, αSMA: green), while lung sections of vehicle-treated mice did not show BiP and αSMA double positive cells (Figure 3A). Moreover, collagen 1 was accumulated around BiP/αSMA double positive cells (Figure 3B, COL-1, cyan). These observations demonstrate that myofibroblasts in lung from Ad-TGFβ mice have increased ER stress and contribute to the deposition of collagen 1 during the development of PF. On the other hand, lung tissue sections from Ad-TGFβ mice treated with CSP showed a notable reduction in BiP/αSMA double positive cells (Figure 3C), indicative of reduced ER stress in LFs. It should be noted that CSP-treated mice revealed a large number of BiP single positive cells that did not show αSMA signal. It should be also noted that TGFβ+CSP-treated mice stained with non-immune IgGs, which matched to the IgG isotype of primary antibodies for BiP, collagen 1, and αSMA, did not detect any fluorescent signals (data not shown). Figure 3 also shows strong signals of αSMA in smooth muscle cells (SMCs) showing either ring- or partial ring-shaped structures of airway and vascular SMCs in lungs. Collectively, these results demonstrated that CSP mitigates the ER stress in myofibroblasts both in vitro and in vivo.

### 2.3. Effect of CSP to Promote Degradation of ECM in hIPF-Lfs

Fibrotic lung tissues show extensive deposition of ECM including collagen 1 that is a critical component of fibrosis [27,28]. In our previous work, we demonstrated that CSP can resolve existing PF in addition to mitigating its progression [9,29]. We hypothesize that the reduction in collagen 1 of the CSP-treated fibrotic lung tissues could be due to its effect on the degradation of collagen 1 by extracellular proteolysis.

To assess the role of CSP on collagen removal in fibrotic lung, we studied the effect of CSP on the capacity of hIPF-Lfs to degrade extracellular matrix components. The degradation activity of hIPF-Lfs was evaluated by using a gelatin degradation assay (Figure 4A). In this assay, fluorescent-labeled gelatin is broken down by secreted proteases such as matrix metalloproteinases (MMPs), and the degraded area can be seen as a dark area by fluorescent microscopy. As shown in Figure 4B, the gelatinolytic activity in TGFβ-treated hIPF-Lfs was significantly enhanced by the treatment with CSP (approximately fourfold higher than that of control), whereas treatment with either TGFβ or CSP alone did not show significant changes in its gelatinolytic activity.

The result suggests the change in proteolytic activity of hIPF-Lfs against ECM proteins. MMPs are known to play an important part in the pathogenesis of pathological tissue fibrosis and the degradation of ECM components [27,28]. Therefore, we examined the effects of CSP on the expression of various MMPs. Expression of mRNA of MMPs were determined by using a quantitative reverse transcription polymerase chain reaction (qRT-PCR) analysis. We examined the expression of MMP 3, 7, 8, 9, 13, 14, 19, and 25 as mice deficient in these MMPs modulate fibrosis, including bleomycin (BML)-induced PF [28,30]. As shown in Figure 4C, among the MMPs tested, MMP13 was markedly upregulated in TGFβ-treated hIPF-Lfs exposed to CSP. These results suggest that MMP13-mediated ECM degradation may contribute to the resolution of IPF by CSP.

### 2.4. Effects of ER Stress Pathways on the Degradation Activity of hIPF-Lfs

UPR pathways consist of three ER transmembrane proteins including PKR-like ER kinase (PERK), activating transcription 6 (ATF6), and IRE1α [16,17,18,19,20,21]. Since we observed that CSP reduced the ER stress in TGFβ-treated hIPF-Lfs through the restoration of the basal level of IRE1α and BiP (Figure 2), we examined whether the IRE1α pathway plays an important role in the gelatinolytic activity of hIPF-Lfs using a specific inhibitor. hIPF-Lfs were incubated with DMSO vehicle or IRE1α inhibitor, 4µ8C. After 48 h incubation, TGFβ-treated hIPF-Lfs with 4µ8C showed significant (*p* < 0.05) elevation of the gelatinolytic activity. This was similar to the administration of CSP in TGFβ-treated hIPF-Lfs (Figure 5A,C). On the other hand, neither the PERK inhibitor, GSK260614, nor ATF6 inhibitors, Ceapin-A7, increased gelatinolytic activity in TGFβ-treated hIPF-Lfs. The increased degradation activity by 4µ8C was unchanged even if it was treated together with either GSK260614 or Ceapin-A7, respectively (Figure 5B,C, *p* < 0.05).

Because the administration of CSP in TGFβ-treated hIPF-Lfs led to a significant increase in MMP13 gene expression and reciprocally reduction in IRE1α protein expression, we examined whether IRE1α signaling pathway regulates MMP13 gene expression. As shown in Figure 6A, administration of 4µ8C significantly elevated the expression of MMP13 mRNA in TGFβ-treated hIPF-Lfs, which is similar to that observed with CSP treatment. We previously showed that TGFβ-mediated induction of collagen 1 expression was reversed in CSP-treated IPF mouse [9]. Therefore, we examined the effect of IRE1 inhibitor on the collagen 1 expression in hIPF-Lfs. Secretion of collagen 1 from TGFβ-treated hIPF-Lfs was significantly reduced by the administration of either 4µ8C or CSP (Figure 6B).

We next asked whether TGFβ increases MMP13 expression in vivo. To address this question, we examined MMP13 expression in the lung from Ad-TGFβ mice treated with CSP. As shown in Figure 7A, we observed robust signals of MMP13 in ameliorated lung from CSP-treated TGFβ mice (Figure 7A, bottom panels: TGFβ+CSP mice), while there was a paucity of signals of MMP13 in lung sections of vehicle and TGFβ-treated mice (Figure 7A, top and middle panels). It should be noted that the lung sections of TGFβ mice showed increased αSMA signals that were relatively reduced in the lung sections from CSP-treated TGFβ mice (Figure 7A).

We next asked whether MMP13 expression is likewise up-regulated in widely used pre-clinical model of PF, i.e., BLM-induced PF model. Lung section from mice exposed to BLM showed elevated αSMA signals that was notably reduced in the lung section of CSP-treated BLM mice (Figure 7B). MMP13 signal of lung sections from BLM-induced PF mice showed slightly increased than control WT mice (Figure 7B, top and middle panels), and robust increase in the signals of MMP13 were observed in ameliorated lung area in CSP-treated BLM mice that was similar to the CSP-treated Ad-TGFβ mice (Figure 7B, bottom panels: BLM + CSP). Interestingly, high magnification image showed that week fluorescent signals of αSMA were also detected in the MMP13 positive cells in mice exposed to BLM with CSP (Figure 7C, surrounded area by dashed lines). Since CSP reduced the expression of αSMA in TGFβ-treated hIPF-Lfs and pre-clinical mouse models of PF [9], these observations suggested that MMP13 positive cells in mouse lung section exposed to BLM with CSP might be derived from BLM-induced myofibroblasts.

## 3. Discussion

IPF is the most common and lethal form of interstitial lung diseases (ILDs) [1,3,31]. It is associated with progressive destruction of lung parenchyma, leading to loss of lung function [3,32]. IPF is increasing in prevalence and the majority of patients present with advanced disease at diagnosis [33]. IPF has a median five-year survival of only 20% and has a mortality rate comparable to or worse than many end-stage malignancies [32]. Until recently, IPF was refractory to all pharmacologic interventions [33]. Lung transplantation is the only viable option for patients with advanced PF [32]. However, recent evidence suggests that pirfenidone and nintedanib can slow the decline of lung function as measured by the endpoint of forced vital capacity (FVC) [5,34]. However, these drugs are not curative and significant side effects or intolerance can occur with their use [35]. Therefore, there is a pressing need for more effective and better tolerated interventions for treatment of patients with IPF or related ILDs. Lung lesions in fibrotic lungs, including IPF lungs, show extensive deposition of ECM, a critical component of alveolar architectural destruction and distortion, leading to progressive loss of respiratory units and lung function [2,3]. Recent studies from our laboratory demonstrated that CSP alleviates the deposition of ECM in multiple preclinical mouse models of BLM- and Ad-TGFβ-induced existing PF [8,9]. Others have proved the beneficial effects of CSP or a related peptide in scleroderma, cardiac, liver, and renal fibrosis [10,11,12,13,14,15]. However, it remains unclear how CSP clears existing ECM. In the current study, we present data that expand current understanding of the mechanisms, by which CSP resolves excess ECM deposited in the lungs.

We examined whether CSP can penetrate hIPF-Lfs and directly interact with intracellular organelles or whether it binds and stays at the membrane surface. Our results revealed that CSP readily internalized into the cells, suggesting that the effect of CSP is not primarily through signaling at the cell surface. Caveolin is a principal component of caveolae that are generally distributed on multiple cell membrane structures [24]. We found the caveolin-derived peptide, CSP, primarily at the ER in hIPF-Lfs and not in other subcellular organelles. Interestingly, our imaging study demonstrated that CSP colocalized with calnexin, an ER-resident chaperone, which localizes in cholesterol-rich microdomains at the mitochondria-associated ER membranes (MAM) [36,37]. Since CSP has a cholesterol-binding motif [24,38], it is plausible that CSP would accumulate at the cholesterol-rich microdomain of ER.

Immunofluorescent imaging of lung tissue revealed that Ad-TGFβ-induced PF generated αSMA/BiP double positive cells in the area near the accumulation of collagen 1 in fibrotic lung lesions. Treatment of Ad-TGFβ-PF mice with CSP attenuated the presence of αSMA/BiP double positive cells in the lung, suggesting that CSP-mediated decreased ER stress of αSMA expressing cells, i.e., myofibroblasts. Lung tissues of Ad-TGFβ mice exposed to CSP, exhibited many BiP single positive cells compared to lung tissues of naïve or Ad-TGFβ-treated control mice. Because BiP is also known to recruit misfolded proteins to ER associated degradation (ERAD) [39,40], the CSP-dependent up-regulation of BiP may play an important role in ERAD to restore cellular homeostasis in the lung cells such as alveolar epithelial cells.

In the present study, we found that CSP reduced TGFβ-induced ER stress via the restoration of the basal level of IRE1α and BiP expression in primary hIPF-Lfs. Halayko and colleagues previously reported that while TGFβ-treated Lfs isolated from patients with or without IPF-induced collagen 1 expression, TGFβ-induced IRE1α/BiP expression was only observed in LFs from patient with IPF [17]. The present results are consistent with the earlier report [17] and further show that CSP modulates ER stress occurring in IPF-Lfs. We infer that these effects could contribute to progression of IPF in vivo.

The beneficial effects and alleviation of collagen 1 accumulation in hIPF-Lfs and IPF lung tissues ex vitro, and in mice with existing PF after CSP treatment are well established [8,9,10]. However, the mechanism of how CSP mitigates collagen 1 deposition in the lung remains obscure. In the present study, we found that CSP enhanced the gelatinolytic activity of hIPF-Lfs exposed to TGFβ, which is associated with a marked increase in MMP13 expression. This finding suggests that CSP reduces the accumulation of collagen 1, in part, through MMP13-mediated digestion of deposited collagen-1. Since the proteolytic activity of MMP9 can be activated by MMP13 [41,42,43], the activated MMP9 could be partially responsible for the clearance of deposited collagen 1 in the lung.

Based on our findings, we believe that CSP alleviates TGFβ-induced PF and could improve lung function, a newly recognized mechanism. CSP mitigates IRE1α expression caused by TGFβ-induced ER stress, which leads to decreased collagen 1 secretion. Our study demonstrated that both CSP-dependent IRE1α suppression and 4µ8C-treated IRE1α inhibition resulted in the reduction in collagen 1 secretion. Supporting this view, it has been reported that ectopic overexpression of X-box binding protein 1 (XBP1), which is the direct downstream target of IRE1α, induced collagen 1 expression in hepatic stellate cells [44]. Since 4µ8C is a selective inhibitor of IRE1α RNase activity, and thus inhibits IRE1α-mediated cleavage of a 26-nucleotides intron from unspliced XBP1 mRNA [45,46,47,48], CSP-dependent suppression of IRE1α may influence the splicing of XBP1 mRNA, which leads to the decrease in collagen 1.

The other is that CSP-dependent IRE1α suppression resulted in the upregulation of MMP13, thus leading a clearance of ECM and fibrin clots. This view is partly supported by a previous report that MMP13 deficient mice are increasingly susceptible to BLM-induced PF [49], while they appeared to resist acute inflammation and fibrosis following exposure to radiation [50]. Further supporting this view, our in vitro and in vivo studies demonstrated that the gelatinolytic activity of TGFβ-treated hIPF-Lfs was enhanced by both CSP-dependent IRE1α suppression and 4µ8C-dependent IRE1α inhibition, and that the results of our immunohistochemistry analyses showed the reduction in collagen 1 and the elevation of MMP13 signals in the CSP-treated PF in two different mice models.

The expression of the MMP13 gene has been known to be regulated by several micro-RNAs that directly modulate MMP13 expression in both negative and positive manners, and that also indirectly control the upstream targets of MMP13 [51,52,53]. Therefore, CSP-dependent ECM degradation may also be facilitated by the expression of MMP13 through protecting microRNA(s) from IRE1α-mediated microRNA decay. Supporting this, it has been reported that some micro-RNAs can be cleaved by the IRE1α-mediated degradation [54]. Understanding of the detailed regulatory mechanisms that control MMP13 expression and activity requires further studies.

## 4. Materials and Methods

### 4.1. Antibodies

Anti-ER stress proteins were analyzed using the ER stress antibody sampler kit (9956, Cell Signaling). Anti-Cellular organelle proteins were assessed using Organelle Maker staining kit (OK7670, ECM Biosciences, Versailles, KY, USA). Goal anti-collagen type 1 antibody was Col-1 (1310-01, SouthernBiotech, Birmingham, AL, USA). Mouse anti-α-tubulin antibody was purchased from cell signaling. Mouse anti-α-actin antibody was purchased Sigma-Aldrich. Rabbit anti-MMP13 antibody was purchased from Novus Biologicals. Non-immune mouse, rabbit, and goat IgGs were purchased from Thermo Fisher Scientific.

### 4.2. Cell Culture

Human Lung Fibroblasts from patients with Idiopathic Pulmonary Fibrosis, (hIPF-Lfs) were purchased from Lonza (Morrisville, NC, USA) and maintained in FGM^TM^-2 Fibroblast Growth Medium-2 BulletKit^TM^ (Lonza). Human IPF Lfs were used at passage 4-7.

### 4.3. Gelatin Degradation

Briefly, 18 mm round glass coverslips were coated with 0.2% FITC-conjugated gelatin and these matrices were crosslinked using 0.5% glutaraldehyde. Cells were incubated with serum-starvation medium (DMEM) for 18 h. Serum-starved cells were seeded on glass coverslips coated with FITC-conjugated gelatin in the presence or absence of CSP (20 µM). After 6 h incubation, cells were completely attached on cover glass and then stimulated with TGFβ (2.5 ng/mL) for 48 h. Cells were fixed and stained with Hoechst 33342 for nuclear and Alexa 568-phalloine for F-actin. Gelatin degradative activity was quantified by measuring the degraded area in cells by use of NIH image program.

### 4.4. Ad-TGFβ- and BLM-Induced Established PF

All experiments using mice were approved by the UTHCT Institutional Animal Care and Use Committee of the UTHSCT (IACUC). Two preclinical murine models of PF, and Ad-TGFβ- and BLM-induced PF, were prepared as described previously [9,55,56].

For the adenovirus expressing constitutively active TGFβ-induced PF model, groups included (*n* = 3 mice per group):A saline control group. Mice were exposed to empty viral vector (Ad-Ev) in saline and, 14 days later, mice were treated daily with saline (200 µL of vehicle) for 14 days.Ad-TGFβ + vehicle group. Mice were exposed to Ad-TGFβ (10^9^ PFU) in saline and, 14 days later, mice were treated daily with saline (200 µL of vehicle) for 14 days.Ad-TGFβ + CSP group. Mice were exposed to Ad-TGFβ (10^9^ PFU) in saline and, 14 days later, mice were treated daily with CSP (200 µL of CSP (1.5 mg/kg)) for 14 days.

At 28 days after Ad-TGFβ transduction, mice were evaluated for changes in PF and tissues were harvested for staining of paraffin-embedded tissue samples [9].

For the bleomycin-induced PF, groups included (*n* = 3–4 mice per group):A saline group. Mice were exposed to saline and, 14 days later, mice were treated daily with saline (200 µL of vehicle) for 7 days.Bleomycin + vehicle group. Mice were exposed to bleomycin and, 14 days later, mice were treated daily with saline (200 µL of vehicle) for 7 days.Bleomycin + CSP group. Mice were exposed to bleomycin and, 14 days later, mice were treated daily with CSP (200 µL of CSP (1.5 mg/kg)) for 7 days.

At 21 days after BLM injury, mice were evaluated for changes in PF and tissues were harvested for staining of paraffin-embedded tissue samples [9,55,56].

### 4.5. Quantitative Real-Time RT-PCR (qRT-PCR)

Total RNA was extracted by using a kit from QIAGEN (Valencia, CA, USA). cDNA was synthesized using SuperScript^TM^ III reverse transcriptase (Invitrogen). The expression levels of specific genes were determined by qRT-PCR using QuantStudio™ 6 Flex Real-Time PCR System (Applied Biosystems, Foster City, CA, USA). TaqMan MGB probes labeled with FAM (Life Technologies, Carlsbad, CA, USA) were used to investigate the expression levels of targeted genes (MMP3, Hs00968305_m1; MMP7, Hs01042796_m1; MMP8, Hs01029057_m1; MMP9, Hs00234579_m1; MMP13, Hs00233992_m1; MMP14, Hs01037003_g1; MMP19, Hs00418247_g1; MMP25, Hs01554789_m1). GAPDH probe labeled with VIC was used as internal reference for normalization. All samples were measured in triplicate.

### 4.6. Western Blot Analysis

Cellular proteins were separated by SDS-PAGE in a 7.5–20% polyacrylamide gradient slab gel [57] and transferred to nitrocellulose membrane. Immunoblotting was performed as previously described [58]. The amounts of proteins either in cell lysates or the culture supernatants were determined by scanning densitometry (BIO-RAD ChemiDoc XRS+ Imaging systems) using NIH image program [59,60].

### 4.7. Immunofluorescence Staining

Immunocytochemistry was performed as previously described [58] with slight modifications. IPF cells were fixed by solution I (4% formaldehyde, 2 mM MgCl_2_, and 1 mM EGTA in PBS) for 10 min and then washed twice with PBS. After extensive washing, the samples were permeabilized with −20 °C cold acetone for 10 min. After permeabilization, the samples were washed three times with PBS and incubated for 30 min with 1% BSA in PBS. Samples were coated with primary antibodies and incubated overnight at 4 °C. The unbound primary antibodies were washed thrice with PBS. The samples were incubated with fluorescence dye conjugated second Abs (Molecular Probes, Inc., Eugene, OR, USA) for 90 min at RT. All the samples, following thrice washes with PBS, were mounted using DABCO solution.

### 4.8. Immunohistochemistry

Sections of 5.0 µm were prepared from representative paraffin blocks of mouse lung samples. Lung immunostaining was performed as previously described [61,62]. Sections were then deparaffinized and rehydrated with xylene and a series grade of alcohol. For IHC staining, heat-induced epitope retrieval was performed by placing the slides in a slide glass container with a citrate buffer (0.1 M citric acid and 0.1 M sodium citrate, pH 6.0) in a water bath at 95 °C for 20 min. After cooling to room temperature, sections were blocked by solution of mouse kit (M.O.M. kit, Vector Laboratories) for 60 min. Then, tissue sections were incubated with a mixture of primary antibodies diluted in blocking buffer at 4 °C for overnight in a humidified chamber. After washing in PBS, tissue sections were incubated for 90 min at room temperature with fluorescent dye-conjugated secondary antibodies and Hoechst 33342 for nuclear staining.

### 4.9. Image Processing

Differential interference contrast (DIC) and fluorescence images were obtained using a Leica TSC SP8 confocal laser scanning microscopy system (Leica Microsystems Inc., Heidelberg, Germany). The optimal settings for the excitation/emission spectrum of given fluorochromes were automatically adjusted by using the Leica SP8 white light laser (WLL) Prism dispersion/Spectral detection system. All images were taken with the same laser output to directly compare with the fluorescence signal intensities [59]. For three-dimensional reconstruction, a series of optical sections were collected either at 0.5 μm intervals in Z-axis (progressively across the cells) or 1.0 µm (progressively across the tissues) intervals in Z-axis, respectively. The images were reconstructed using LCS 3D software of Leica Microsystems [59,60].

### 4.10. Statistical Analysis

Data were expressed as mean ± standard error of the mean (SEM). Statistical significance was tested with a one-way ANOVA followed by the Dunnett’s test for comparing between two groups, a one-way ANOVA followed by the Tukey’s test for comparing among multiple groups. Differences were considered significant if *p* was < 0.05.

## 5. Conclusions

In summary, we uncovered dual regulatory roles of CSP that may contribute to the resolution of established PF. Our findings support the concept that CSP may facilitate the restoration of lung function by reducing the secretion of ECM to prevent expansion of lesions and inducing the expression of MMP13 to clear existing ECM deposits in pulmonary fibrosis.

## Figures and Tables

**Figure 1 ijms-23-03316-f001:**
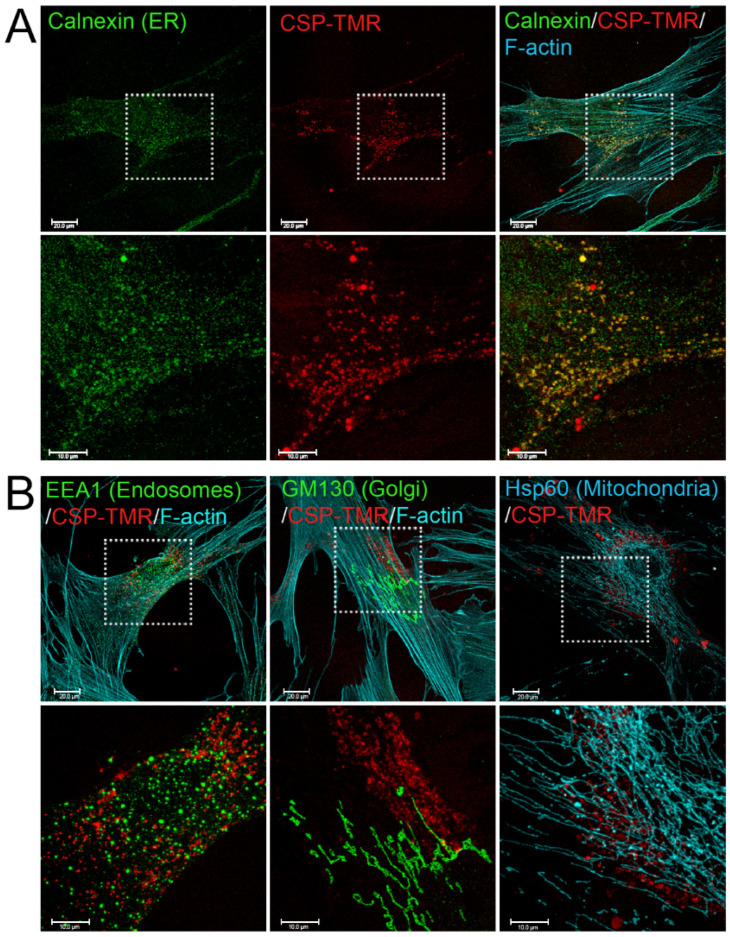
Internalization and Subcellular Distribution of fluorescent-labeled CSP in human IPF-Lfs. hIPF-Lfs were treated with tetramethylrhodamine-conjugated CSP (CSP-TMR, 10 µM) for 2 days and then immunostained with organelle makers such as Calnexin for endoplasmic reticulum (ER), EEA1 for early endosomes, GM130 for Golgi, and Hsp60 for Mitochondria. Lower panels in (**A**,**B**) show higher magnification of the boxed areas of the upper panels. Scale bars are 20.0 µm (lower magnification images) and 10.0 µm (higher magnification images), respectively.

**Figure 2 ijms-23-03316-f002:**
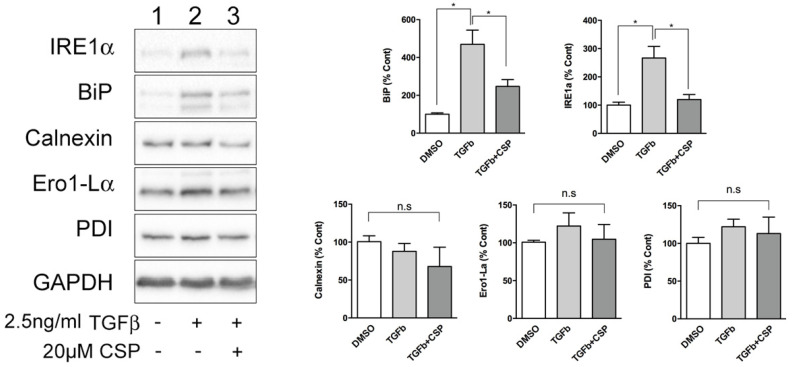
CSP reduced TGFβ-induced ER stress in hIPF-Lfs. Expression of unfolded protein response (UPR) markers in hIPF-Lfs. hIPF-Lfs were treated with TGFβ in the presence or absence of CSP. Whole cell lysates were subjected to Western blotting for IRE1α, BiP, calnexin, Ero1-Lα, PDI, and αSMA. The expression of the UPR markers was normalized to that of glyceraldehyde 3-phosphate dehydrogenase (GAPDH) and is shown as mean ± SEM from three independent experiments. GAPDH was used as an internal control to normalize protein expression. * *p* values by one-way ANOVA with the Tukey’s multiple comparisons test (*p* < 0.05).

**Figure 3 ijms-23-03316-f003:**
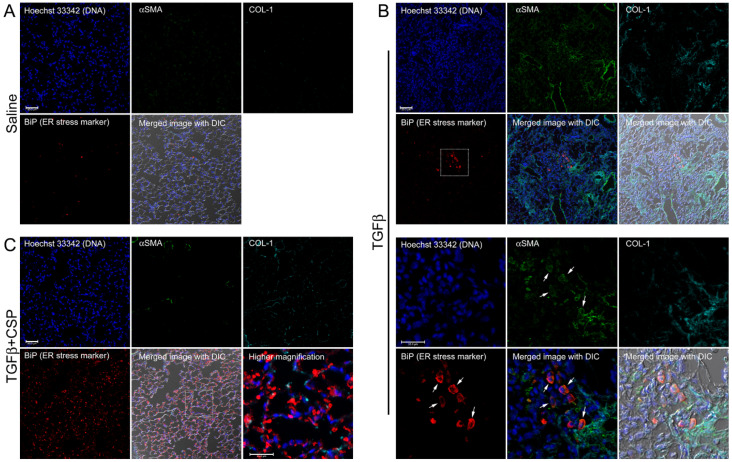
Effect of CSP on ER stress in fibrotic lung fibroblasts in mouse PF lung. Lung sections from mice exposed to (**A**) Saline, (**B**) Ad-TGFβ, and (**C**) Ad-TGFβ+CSP were stained with DNA (blue), BiP (ER stress marker) (red), collagen 1 (cyan), and αSMA (fibrotic fibroblasts marker) (green). Lower panels in (**B**) show higher magnification images of the upper white boxed area. Arrows show BiP and αSMA double positive cells. Right bottom corner in (**C**) shows a higher magnification image of the white boxed area. Scale bars are 50.0 µm (lower magnification images) and 25.0 µm (higher magnification images), respectively. Images are representative of 10 fields/slide (*n* = 3 mice per group).

**Figure 4 ijms-23-03316-f004:**
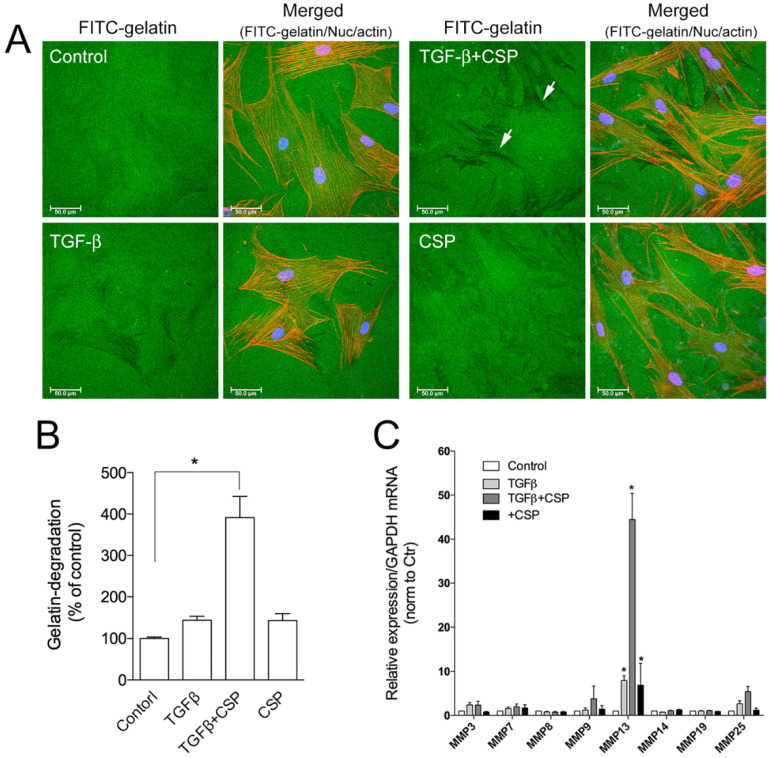
Effect of CSP on gelatin-degradation activity in hIPF-Lfs: (**A**) CSP enhances gelatin-degradation activity. Forty-eight hours after TGFβ (2.5 ng/mL) stimulation in the presence or absence of 20 µM CSP, hIPF-Lfs were seeded on glass coverslips coated with FITC-conjugated gelatin. Twenty-one hours later, cells were fixed and stained with nuclear (blue) and F-actin (red). Scale bars: 50 µm. Arrows indicate degradation areas. (**B**) The gelatin degradation was quantified by measuring the degraded area of the FITC-conjugated gelatin using the NIH image program. Values are means ± SEM from three independent experiments. The degradation area of serum-starved cells was determined as a control of reference. *p* * values by one-way ANOVA with the Dunnett’s multiple comparisons test (*p* < 0.05). (**C**) Effect of CSP on MMPs gene expression. TGFβ-stimulated hIPF-Lfs were treated either with or without CSP (20 µM). cDNA from total RNA in each condition was subjected to qPCR to determine gene expression level of MMPs. Values are means ± SEM from three independent experiments. GAPDH mRNA levels were used as an internal control to normalize gene expression. * *p* values by one-way ANOVA with the Tukey’s multiple comparisons test (*p* < 0.05).

**Figure 5 ijms-23-03316-f005:**
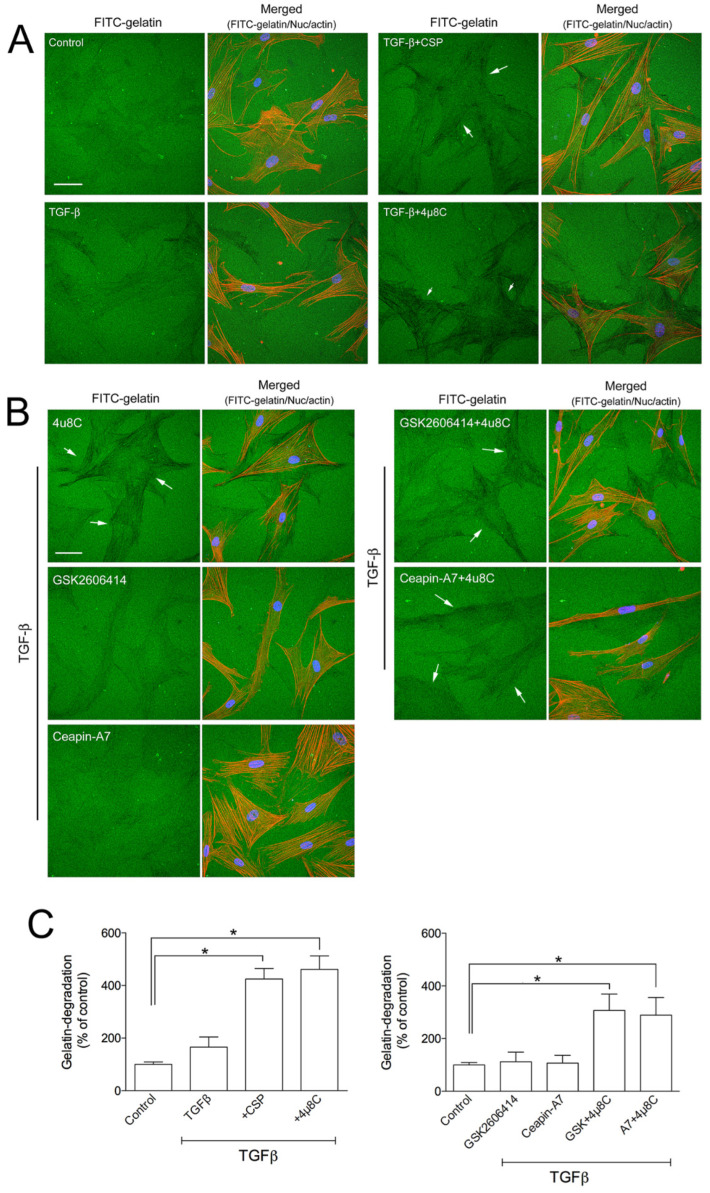
Effects of ER stress inhibitors on hIPF-Lfs gelatin degradation activity. hIPF-Lfs were seeded on glass coverslips coated with FITC-conjugated gelatin (green) under serum-starved conditions. After 18 h incubation, cells were treated with (**A**) CSP (20 µM), 4µC8 (25 µM), (**B**) 4µC8 (25 µM), GSK260641 (1 µM), and Ceapin-A (10 µM) in the presence or absence of TGFβ for 48 h. These cells were then fixed and stained with nuclear (blue) and F-actin (red). Scale bars: 50 µm. Arrows indicate degradation areas. (**C**) The gelatin degradation was quantified by measuring the degraded areas using the NIH image program. Values are means ± SEM from three independent experiments. The degradation area of serum-starved cells was determined as a control of reference. *p* * values by one-way ANOVA with the Dunnett’s multiple comparisons test (*p* < 0.05).

**Figure 6 ijms-23-03316-f006:**
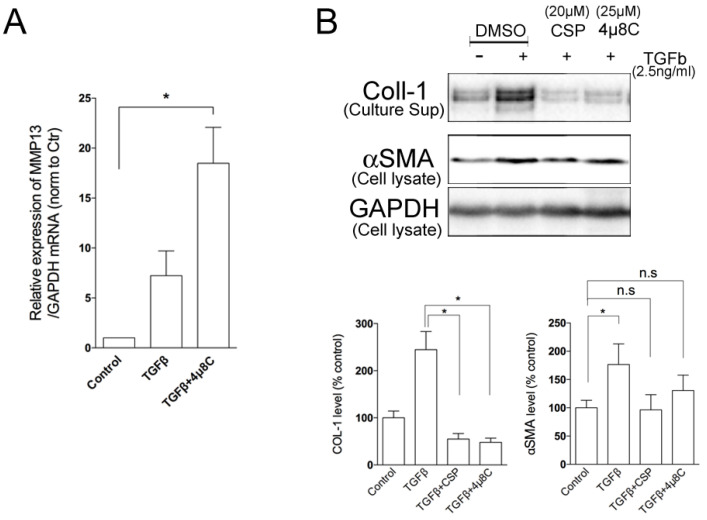
Effects of IRE1α inhibitor on MMP13 and Collagen 1 secretions. TGFβ-stimulated human IPF LFs were treated either with or without 4µ8C (25 µM). (**A**) IRE1α inhibitor enhanced the expression of MMP13 in hIPF-Lfs. cDNA synthesized from total RNA in each condition was subjected to qPCR to determine gene expression level of MMP13. Values are means ± SEM from three independent experiments. GAPDH mRNA levels were used as an internal control to normalize gene expression. *p* * values by one-way ANOVA with the Dunnett’s multiple comparisons test (*p* < 0.05). (**B**) An IRE1α inhibitor, 4µ8C, reduced the secretion of collagen 1 from human IPF LFs. Culture supernatants and whole cell lysates were subjected to Western blotting for collagen 1 (COL-1). Upper panel: Representative Western blot images. Lower Panels: Statistical representation of the Western blot analysis. Values are means ± SEM from three independent experiments. GAPDH was used as an internal control to normalize protein expression. *p* * values by one-way ANOVA with the Dunnett’s multiple comparisons test (*p* < 0.05).

**Figure 7 ijms-23-03316-f007:**
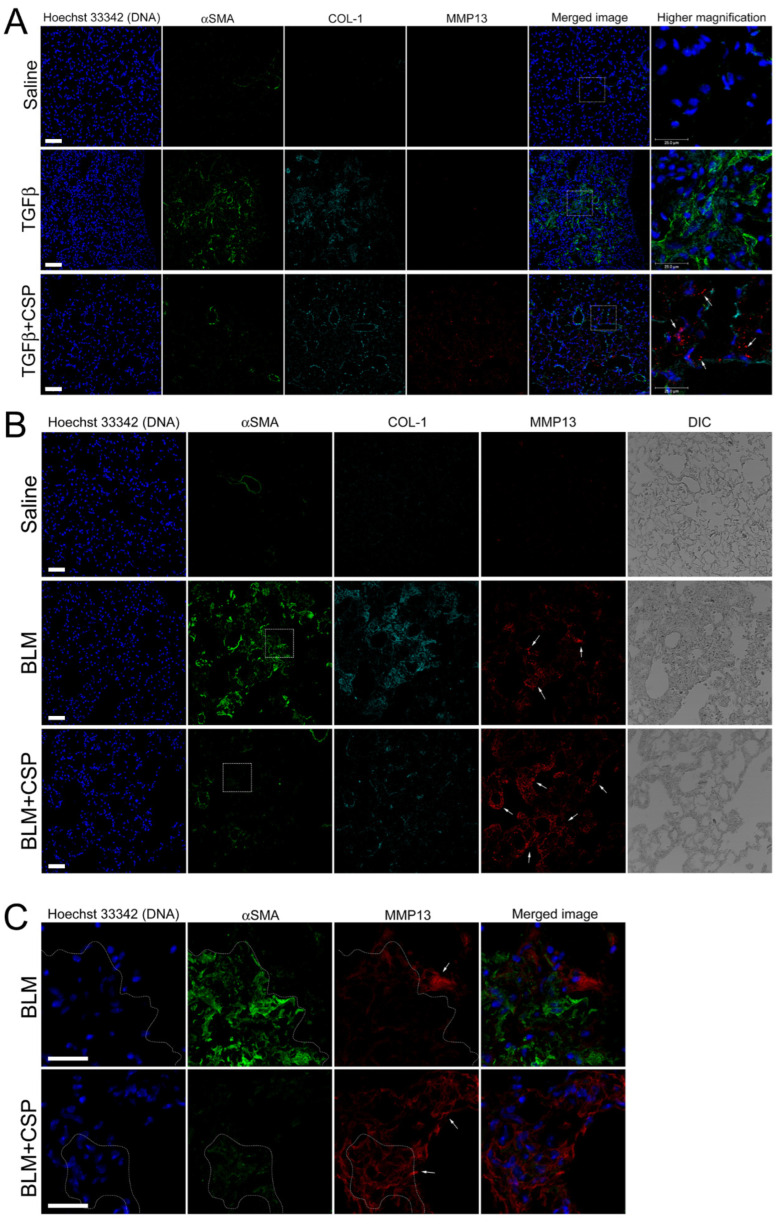
CSP promoted MMP13 expression in murine models of lung fibrosis: (**A**) Effect of CSP on TGFβ-induced established PF. Lung sections from mice exposed to Saline, Ad-TGFβ, and Ad-TGFβ+CSP were stained with DNA (blue), MMP13 (red), collagen-1 (cyan), and αSMA (fibrotic fibroblasts marker) (green). Scale bars: 50 µm. Right panels show higher magnification images in the white boxed areas. Arrows indicate MM13. Bars: 25 µm. Images are representative of 10 fields/slide (*n* = 3 mice per group). (**B**,**C**) Expression of MMP13 in BLM-induced PF in mice. Lung sections from mice exposed to Saline, BLM, and BLM+CSP were stained with DNA (blue), MMP13 (red), collagen-1 (cyan), and αSMA (fibrotic fibroblasts marker) (green). Scale bars: 50 µm. (**C**) Higher magnification images in either BLM or BLM + CSP in panel A (white boxed areas). Scale bar 25 µm. Images are representative of 10 fields/slide (*n* = 3–4 mice per group).

## Data Availability

Not applicable.

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
