# Peer review of "Caveolin-1-Derived Peptide Reduces ER Stress and Enhances Gelatinolytic Activity in IPF Fibroblasts"

_ijms, 2022, doi:10.3390/ijms23063316_

Round 1
Reviewer 1 Report
In this manuscript, the author reports, ‘Caveolin-1 derived peptide reduces ER stress and enhances gelatinolytic activity in IPF Fibroblasts’. The authors should address the following questions before getting a possible publication.
Recommendation: Minor revisions needed as noted.
- The novelty of the present work should be discussed in the Introduction section.
- The manuscript contains error bars in several figures. What does the error bars stand for ? It should be mentioned in the Figure captions.
- Which excitation filter was used for the fluorescence images in Fig.1?
- The author should write the purpose for each test in one/two sentences (in brief) before explaining the results of the characterization techniques. Therefore, the logic and organization of this part will be enhanced.
- The authors should cite some updated relevant references in the Introduction section.
- The formatting and grammatical errors in the article need to be checked carefully.
Author Response
Response to review’s 1
Comments and Suggestions for Authors
In this manuscript, the author reports, ‘Caveolin-1 derived peptide reduces ER stress and enhances gelatinolytic activity in IPF Fibroblasts’. The authors should address the following questions before getting a possible publication.
Recommendation: Minor revisions needed as noted.
- The novelty of the present work should be discussed in the Introduction section.
Response:We thank for the valuable comments. According to the reviewer’s recommendation, we discussed the novelty of the present work in new “Introduction” in the revised manuscript. The changes in manuscript are in red font.
- The manuscript contains error bars in several figures. What does the error bars stand for ? It should be mentioned in the Figure captions.
Response: We apologize for the insufficient statement. To address the reviewer’s suggestion, we added following statement in the revised manuscript: “GAPDH was used as an internal control to normalize protein expression” for Western blots and “GAPDH mRNA levels were used as an internal control to normalize gene expression” for qPCR in the revised manuscript. We also mentioned that “the degradation area of serum-starved cells was determined as a control reference” in the revised manuscript. Error bars are Standard Error of the Mean (SEM).
- Which excitation filter was used for the fluorescence images in Fig.1?
Response:The tuning range of Leica SP8 white light laser (WLL) confocal system is 470 to 670 nm in 1 nm intervals. The excitation/emission spectrum profileof fluorochromesare installed in the Leica SP8 WLL confocal system and the optimal settings forthe excitation/emission spectrum of given fluorochromes are able to adjust automatically. We also used a sequential scan mode to minimize crosstalk in multiply stained specimens, including autofluorescence signals.
In Figure 1, the excitation for Alexa fluor-488 dye was 499 nm and the emission (Prism range) was 505-556 nm. The excitation for TRITC-CSP was 551 nm and the emission (Prism range) was 556-631 nm, and the excitation and emission for Alexa fluor-647 dye were 653 nm and 658-779 nm, respectively. In the revised manuscript, we have included following statement: “The optimal settings forthe excitation/emission spectrum of given fluorochromes were automatically adjusted by using the Leica SP8 WILL Prism dispersion/Spectral detection system” in the methods section.
- The author should write the purpose for each test in one/two sentences (in brief) before explaining the results of the characterization techniques. Therefore, the logic and organization of this part will be enhanced.
Response: We thank for the valuable comments. According to the reviewer’s suggestion, the purpose for each experiment is clarified in the revised manuscript.
- The authors should cite some updated relevant references in the Introduction section.
Response:According to the reviewer’s suggestion, we have cited new references in new Introduction section.
- The formatting and grammatical errors in the article need to be checked carefully.
Response:We apologize for the formatting and grammatical errors.The revised manuscript has been carefully edited to rectify such errors.
Reviewer 2 Report
The authors investigated about the mechanism by which CSP protects against pulmonary fibrosis.The rational behind the study was clear and straight forward. The manuscript is almost well written.
Overall the topic could be interesting but many details are not clear.
I recommend that the paper be accepted with minor revision:
a). The authors should mentioned in the abstract more details about model used.
b) In the introduction section, little previous evidence is provided about the importance of IPF in daily life. Incorporating comparisons with other studies would increase the strength of the paper. Please refer to doi: 10.3390/antiox9070601; 10.1016/j.matbio.2018.03.015; 10.3390/ijms21207761.
c) The authors should add the number of mice used in their study and how they choose the number.
d) The authors should better emphasize the conclusions.
e) There are some minor grammar issues that should be fixed in order to aid the accessibility of the results to the reader.
- b) The authors should list experimental groups for easier understanding in the materials and methods section.
Author Response
Response to review’s 2
Comments and Suggestions for Authors
The authors investigated about the mechanism by which CSP protects against pulmonary fibrosis. The rational behind the study was clear and straight forward. The manuscript is almost well written. Overall the topic could be interesting but many details are not clear.
I recommend that the paper be accepted with minor revision:
a). The authors should mention in the abstract more details about model used.
Response:We thank the reviewer for valuable comments. To address this concern, we described the two preclinical murinemodels of PF that were used in the “Abstract” of the revised manuscript. The changes in manuscript are in red font.
- b) In the introduction section, little previous evidence is provided about the importance of IPF in daily life. Incorporating comparisons with other studies would increase the strength of the paper. Please refer to doi:10.3390/antiox9070601;10.1016/j.matbio.2018.03.015; 10.3390/ijms21207761.
Response: Per the reviewer’s suggestion, we have described the treatment of patients with IPF in greater detail and also mentioned the potential application of CSP for other forms of tissue fibrosis. The revised manuscript also now cites the recommended reference.
- c) The authors should add the number of mice used in their study and how they choose the number.
Response:We apologize for the confusion of mice groups of FP. The numbers of mice used in the different groups were based on power analyses used in several of our previous publications(1-3), which is now clarified in the Figure legends and the materials and methods section in the revised manuscript.
- 1. S. Marudamuthuet al., Caveolin-1-derived peptide limits development of pulmonary fibrosis. Sci Transl Med11(2019).
- Y. P. Bhandaryet al., Regulation of lung injury and fibrosis by p53-mediated changes in urokinase and plasminogen activator inhibitor-1. Am J Pathol183, 131-143 (2013).
- Y. P. Bhandaryet al., Regulation of alveolar epithelial cell apoptosis and pulmonary fibrosis by coordinate expression of components of the fibrinolytic system. Am J Physiol Lung Cell Mol Physiol302, L463-473 (2012).
- d) The authors should better emphasize the conclusions.
Response:Thanks for your kind comment. According to the reviewer’s suggestion, we have prepared “CONCLUSIONS” as a dedicated section to address this point.
- e) There are some minor grammar issues that should be fixed in order to aid the accessibility of the results to the reader.
Response:We apologize for the formatting and grammatical errors.The revised manuscript has been carefully edited to rectify such errors.
- b) The authors should list experimental groups for easier understanding in the materials and methods section.
Response:We apologize for the confusion of mice groups of FP. According to the reviewer’s suggestion, we have described list of experimental groups in the Materials and Methods section in the revised manuscript.